# Impact of an Active Learning Methodology on Students' Emotions and Self-Efficacy Beliefs towards the Learning of Chemical Reactions—The Case of Secondary Education Students

**Maria Antonia Dávila-Acedo** [1], **Jesús Sánchez-Martín** [1], **Diego Airado-Rodríguez** [2] **and Florentina Cañada-Cañada** [1,*]

[1] Departamento de Didáctica de las Ciencias Experimentales y Matemáticas, Universidad de Extremadura, 06006 Badajoz, Spain; mdavilaacedo@unex.es (M.A.D.-A.); jsanmar@unex.es (J.S.-M.)
[2] Department of Science Education, Universidad de Jaén, 23071 Jaén, Spain; dairado@ujaen.es
* Correspondence: flori@unex.es

**Abstract:** The main goal of this article was to analyse the impact of the implementation of an active methodology, namely inquiry-based learning, on the emotions experienced by students and their self-efficacy beliefs toward the curricular content in relation to chemical reactions. A sample of 56 students at the third level of compulsory secondary education (K-9) was considered. It was found that the use of an active methodology had on the one hand, a statistically significant effect on the promotion of positive emotions and a decrease in negative emotions and on the other hand a positive effect on the improvement of students' self-efficacy beliefs towards learning the curricular content related to chemical reactions. The implementation of practical activities in the classroom might lead to better learning of scientific content; at least, the results presented in the current article point in that direction.

**Keywords:** emotions; self-efficacy beliefs; chemical reactions; active learning

## 1. Introduction

The impact of the affective and cognitive dimensions on the teaching–learning process has been recognised in the literature [1–6]. It has been stated that emotions play a key role in the whole teaching and learning process of all disciplines, including, of course, scientific disciplines [7,8].

Tobin (2012) [9] and Tomas and Ritchie (2012) [10] established that emotions are at the core of the process of science learning, since according to these authors, emotions act as a kind of social glue interconnecting both collective and individual actions and interests. The connection of emotions to actions, decision-making [11] and academic success [12,13], has also been stated in the literature. This becomes particularly important for students at the end of their secondary education stage when they have to decide on the direction of their higher degree studies. From the teachers' perspective, knowing about the emotions of their students towards any discipline such as Physics and Chemistry, would help them to reflect and effectively plan their teaching action for significant learning [14].

Efficient learning of science constitutes a complex cognitive process that implies the ability to do and to want to do, in addition to receiving an instruction [15]. It has been stated by Bisquerra (2009) [16] that the teaching–learning process is more fruitful when students have emotional competencies. This is a must in relation to the analysis of not only the cognitive aspects, but also the affective dimension towards the learning of the content of the science curriculum [17]. In this line, the monitorisation of the emotions experienced daily in compulsory education classrooms will serve as a basis for the advanced planning of science interventions, through the design of tasks that tend to produce more positive

emotions [18,19]. This is important since it has been stated that positive emotions promote learning while negative emotions restrict the student's disposition for learning [20,21].

Emotions and feelings have traditionally been classified according to different criteria, which have been derived from multiple taxonomies [8]. However, when studying the effect of emotions on behaviour [16], it is a good approach to divide them in two categories: positive and negative emotions. Moreover, many definitions have been proposed for the concept of emotion, but in this article, emotions will be considered as complex processes in connection with subjective reactions to a personal event or situation involving both behavioural and physiological shifts [22].

In the case of compulsory secondary education students, i.e., adolescents, the appearance of positive emotions toward scientific curricular content is linked to a student's self-efficacy, which is the belief in their capacity to assimilate those contents. Self-efficacy is closely connected to self-regulation, and its capacity for predicting a student's achievement has been demonstrated [23]. Borrachero, Brígido, Mellado, Costillo and Mellado (2014) [24] analysed, with a sample of 510 undergraduates, the affective memory towards science. When the study was performed, these undergraduates were taking different university degrees and they were asked about their emotions during their compulsory secondary education stage. It was concluded that positive emotions toward a certain content increased with the believe of the student in his/her own capacity to learn that content. In the same line, negative emotions increased when they did not consider themselves able to learn certain content. This becomes especially important for the contents of the Physics and Chemistry curricula since more incidences of negative emotions were detected with a significant influence of self-efficacy [25].

Self-efficacy has been defined from the Social Cognitive Theory as a subjective judgement of the level of competence of a person for presenting certain behaviours or reaching certain outcomes in the future. In this line, self-efficacy is in close connection with the self-confidence to believe that what can be done has to be done [26]. In the academic context, self-efficacy is associated with a student's performance and their potential to expand their academic and professional interests, goals and motivation [25,27].

In addition, self-efficacy is in close connection with self-regulation as well as self-thoughts, feelings and actions to achieve our learning goals [28–30]. Thus, the students with high capacities and expectations could face complex challenges as a way of overcoming them and they express positive emotions such as pride, pleasure, fun and joy. On the other hand, students with low capacities and expectations experience negative emotions as anxiety or frustration because they are not able to solve a task [31,32].

Secondary education students could construct an impression of school science as difficult, boring, of little real application, highly disconnected from the real world and overly theoretical. This image will irremediably cause an attitudinal decrease and the appearance of negative emotions towards the sciences. Other causes of this problem could be the influence of the instructor, the lack of hands-on activities or the excessive orientation of the classes to exam preparation [33].

Wan and Lee (2017) [34] have recently recognised that more research is needed on the causal relation between the cognitive and affective domains in secondary education students, especially when dealing with the content of the science curriculum.

Teaching methodology is a key issue in the acquisition of knowledge [35,36]. The employed methodology is also often related to the emotion, motivation and self-regulation of learning [37,38].

Students' motivational aspects constitute one of the current educational challenges faced by teachers. Traditionally it has been recognised that the interest in motivation is a way of improving the performance of the teaching–learning process [39]. In the related literature, initiatives for changing the way the students face up to the acquisition of the content of the science curriculum, have been reported. In this line, for instance, gamification [40], the flipped classroom [41] or guided inquiry-based learning activities [42] have been reported as efficient methodologies for science learning, with a subsequent

positive impact on the affective dimension. Experiences based on active methodologies have lately been reviewed, with emphasis on their impact on learning success and on encouraging the learning process [43].

Therefore, it is necessary to apply strategies focused on active learning to achieve a cognitive and emotional change in students. In this context, experimental activities could be considered as a good option. Hands-on activities involve observation and an understanding of the phenomena, testing of the hypotheses, the implementation of strategies and research development skills in problem solving [44] and an improvement in self-efficacy beliefs [45] making use of a range of methods and strategies.

In this article the impact of the implementation of an active learning methodology based on hands-on activities on emotions and self-efficacy is studied. Chemical reactions were the considered curricular content as a case study. A comparative study was carried out considering a control group, in which a traditional methodology was employed and an experimental group, in which the active methodology based on hands-on activities related to physical and chemical changes was implemented. Figure 1 shows a scheme of the considered methodological approach.

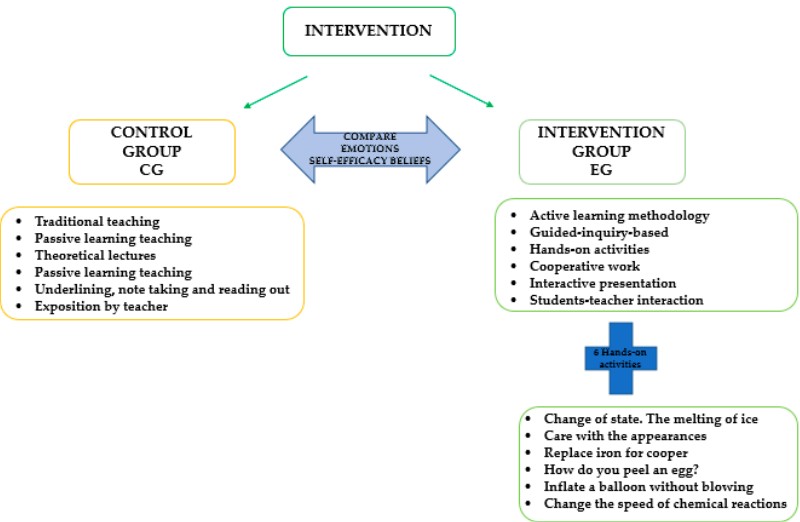

**Figure 1.** Scheme of the applied methodological setting.

## 2. Material and Methods

### 2.1. Sample

The considered sample consisted of 56 students of 3rd year compulsory secondary education, i.e., K-9 students according to international standards, (ages ranged: 14–15 years old) of a secondary education school in Badajoz (Spain). The students in the sample were randomly divided into two groups. In the first group (control group—CGA: $n = 28$), a traditional teaching methodology was employed, while in the second group (experimental group—EGB: $n = 28$), an active learning methodology based on hands-on activities was implemented. Table 1 shows the distribution of the sample by gender.

**Table 1.** Sample distribution according to the gender.

|  |  | Control Group A | | Experimental Group B | |
| --- | --- | --- | --- | --- | --- |
|  |  | *n* | % | *n* | % |
| Gender | Men | 18 | 64.3 | 13 | 46.4 |
|  | Women | 10 | 35.7 | 15 | 53.6 |

### 2.2. Procedure

The designed methodological approach was implemented over four weeks in the month of March. The comparison between the two groups of students was based on the

design of an intervention plan that involved emotions and self-efficacy progression during the learning of the curricular content related to physical and chemical changes.

Traditional magistral expositive lessons were employed with the control group (28 students): The teacher explained the concepts of the physical and chemical changes and the students took down their notes, underlined them and read them out.

The experimental group consisted of 28 students and in this case, an innovative teaching strategy by guided-based-learning (inquiry-based-learning) was employed, including the development of a didactic unit on chemical reactions that was designed with up to 6 different hands-on activities. Each hands-on activity was focused on some aspect of the curricular content related to chemical reactions, including experiments with everyday material systems.

This active and innovative strategy is a way of teaching science based on the experience of the students, generating in them the need to acquire new knowledge during the development of educational intervention. The promotion of the cooperative participation of the students was planned. The students were divided into groups and each group was provided with different materials for daily use. It is important to note that these students had no experience with this teaching strategy and they had never experienced hands-on activities in their schooling stage before. The didactic intervention was carried out in three sessions of 60 min.

The six experimental hands-on activities (Figure 1) were designed to work on the following contents:

Physical and chemical changes;
Chemical reactions. Equations and equation balance;
Types of chemical reactions;
Speed of chemical reactions;
The law of conservation of mass.

Each of the sessions is described below:

The first experience was carried out in the first session. An ice cube was taken, left on the table and after some time it was observed. At the same time, the process of melting butter was carried out. Then, the students were asked to answer different questions proposed by the teacher related to the states of aggregation and the changes of state of matter. Next, the teacher introduced the concept of physical change. Then, the following experiences were carried out: change of state; the melting of ice (activity 1); care with the appearances (activity 2); mixture of substances; separation of the mixture into its components (activity 3).

In the second session, the teacher proposed a problem. An apple was taken, opened in half and left on the table. After a few minutes, it was observed that the apple changed its appearance. At that time, the teacher introduced the chemical reaction concept; she explained the oxidation reaction and she described its presence in many situations of daily life. Moreover, it was explained that this reaction takes place not only in food but also in metal elements that change colour as time passes as a consequence of oxidation. The following experiences were carried out: iron replaced with copper via redox process (activity 4); how do you peel an egg? (activity 5); how do you inflate a balloon without blowing? (Activity 6).

In the third session, an apple was taken by the teacher and cut into three parts. The first part was left on the table, the second one was wrapped with kitchen plastic and the third one was given the juice of a lemon. After a few minutes, the three pieces of apple were observed. It was detected that part number one had changed its appearance; the change in the appearance of part number two was minor; and in part number three there was no change of colour. Then, the teacher introduced and explained the concept of the speed of chemical reactions, the factors that influence it (other examples of daily life) and the role of catalysts. The following experiences were carried out: modification of the speed of chemical reactions (activity 7).

*2.3. Instrument*

A questionnaire was employed as an instrument for data collection. The employed questionnaire was designed and validated previously [42]. Students were asked to anonymously and voluntarily fill-in the questionnaire.

The questionnaire consisted of three sections and it was structured as follows:

1. The first section consisted of list of a personal variables to obtain information about the sample intended to collect student's descriptive information, such as age, gender and academic performance.

2. The second section monitored the emotions experienced by students when receiving a traditional intervention or an active learning one. Eleven emotions were considered, including both positive and negative emotions [42]. In this section, the students were asked to grade the emotions experienced when learning about physical and chemical changes on an 11-point Likert scale from 0 (the lowest incidence) to 10 (the highest incidence) [24,46]. In addition to scoring the experienced emotions, students were also asked to explain the given score for each one.

The internal consistency of the questionnaire was verified through the calculation of its reliability using the covariation among items of the different scales. All the obtained values were above 0.80, as reflected in Table 2, which means that the questionnaire fulfilled the reliability and internal consistency requirements.

**Table 2.** Classification of the considered emotions and the internal consistency of the questionnaire.

| Classification of Emotions | | | |
|---|---|---|---|
| Positive | Joy<br>Trust<br>Fun<br>Enthusiasm<br>Satisfaction<br>Tranquility | Negative | Boredom<br>Fear<br>Nervousness<br>Worry<br><br>Sadness |
| $\alpha$ | 0.833 | $\alpha$ | 0.820 |
| $N$ | 6 | $N$ | 5 |

3. The third section of the instrument consisted of eleven items, was worded in the form of affirmations and aimed to measure the progress of students' self-efficacy when learning the curricular content related to chemical reactions. In this part of the questionnaire, students were requested to score each item using a 0 to 10 scale (0 being the lowest and 10 the highest). These closed-ended questions were of the author' own elaboration, based on the questionnaire of Thomas et al., (2008) [47], The SMELIES: Development of an instrument Designed to Investigate Elements of Science Students' Metacognition, Self-Efficacy and Learning Processes. Table 3 contains the list of the eleven items of self-efficacy considered.

**Table 3.** List of considered self-efficacy items.

| Code | Self-Efficacy Item |
|---|---|
| I1 | I can get good results in this topic |
| I2 | I have been able to relate learned concepts about chemical reactions to aspects of daily life |
| I3 | I have been able to understand the chemical reaction concept |
| I4 | I have been able to understand law of mass conservation in the simple processes |
| I5 | I have been able to understand the concepts about physical and chemical change |
| I6 | I have been able to identify a chemical change |
| I7 | I have been able to identify a physical change |
| I8 | I have been able to distinguish chemical change from a physical change |
| I9 | I have been able to read, represent and adjust chemical reactions |
| I10 | If I try hard enough, I will be able to solve tasks |
| I11 | I have been able to relate that I have learned about chemical reactions with hands-on activities outside the classroom |

*2.4. Data Analysis*

Students employed an average time of 30 min to complete the questionnaire and they were implemented immediately after the third session. Filled-in questionnaires were collected and the data were processed and analysed. Firstly, a descriptive analysis was performed to describe and to establish conclusions. The sample had previously been tested and was found to be homogeneous. Then, the Kolmogorov–Smirnov normality test ($p > 0.050$) was applied and it was found that the sample fulfilled the requirements of normal distribution. In this analysis, a confidence level of 95% was considered.

Then, the *t*-Student test was applied for a comparison of the mean values and to statistically assess the differences between the control and the experimental group. For this statistical analysis, the statistical package SPSS (Statistical Product and Service Solutions) 22.0 was used.

Analysis of the data about emotions and self-efficacy was also performed by means of principal component analysis (PCA) in order to detect the existence of correlations among the variables.

PCA is a well-known technique for dimensionality reduction through the maximization of the variance of a linear combination of the variables. Applying PCA on a given dataset supposes the mapping of original data from the original *d*-dimensional space to a new *k*-dimensional space being $k < d$. The new generated *k* dimensions are known as principal components (PC). Each PC points in the direction that retains the maximum variance of the original dataset, excluding the variance already captured by the preceding PCs. Thus, the first PC is the direction of the new *k*-dimensional space capturing the maximum variance of the original dataset; the second PC is the direction with maximum variance in an orthogonal direction to the first PC; and so on. The use of PCA for the analysis of the data related to emotions is supported by the literature [48–50].

**3. Results and Discussion**

*3.1. Construct Validation and Reliability of the Emotion Scale (Positive and Negative)—Principal Component Analysis of the Emotion's Dataset*

Analysis of the collected data about emotions experienced by compulsory secondary education students (K-9) towards the study of the curricular contents of Physics and Chemistry was carried out by means of PCA. Application of PCA allows significant PCs that summarise the complex information contained in the original dataset to be extracted for easy data interpretation. In first place, to check the suitability of the considered sample to be analysed by PCA, the KMO (Kaiser–Mayer–Olkin) index was verified and the original dataset was also analysed with Bartlett's sphericity test. A KMO index of 0.851 (high≈1) was obtained and a *p*-value < 0.05 was obtained in the Bartlett's sphericity test. The obtained results for both parameters indicated that the PCA could be efficiently applied in the current dataset, considering the eleven monitored emotions (six positive and five negative emotions).

The application of the PCA revealed that the set of considered emotions experienced by the K-9 students towards the study of chemical reactions could be represented by two factors or principal components, one of them representing positive emotions and the other representing the negative ones. The results of the factorial analysis are summarised in Table 4. The two obtained principal components explained 65.35% of the total variance of the original dataset.

Table 4 shows the results of the factorial analysis for emotions and these results assisted in the interpretation of the correlations among the variables (emotions in this case). The emotions of joy, trust, fun, enthusiasm and satisfaction possessed high loading values for the first component, which meant that this factor or component represented positive emotions and all of them were positively correlated. On the other hand, the emotions of boredom, fear, nervousness, worry and sadness had high values of loading for the second component, which indicated that this factor or component represented negative emotions and that the

considered negative emotions were also positively correlated among them. At the same time, both groups of emotions, positive and negative ones, were negatively correlated.

**Table 4.** Results of the factorial analysis of emotions towards the learning of chemical reactions.

| Emotions | First Component | Second Component |
|---|---|---|
| Joy | 0.881 | |
| Trust | 0.828 | |
| Fun | 0.762 | |
| Enthusiasm | 0.808 | |
| Satisfaction | 0.789 | |
| Boredom | | 0.698 |
| Fear | | 0.852 |
| Nervousness | | 0.735 |
| Worry | | 0.745 |
| Sadness | | 0.708 |

*3.2. Construct Validation and Reliability of the Self-Efficacy Scale—Principal Component Analysis of the Self-Efficacy Dataset*

Data collected about self-efficacy were also analysed by means of a PCA in order to check if the considered self-efficacy items could be grouped into a small number of categories or factors in an attempt to summarise the information contained in the original dataset. The PCA also allowed the existence of correlations among the considered self-efficacy items to be detected, which made the interpretation of the data easier. In first place, the suitability of the self-efficacy dataset to be analysed by the PCA was checked. With this purpose, the KMO (Kaiser–Mayer–Olkin) index was estimated and the Bartlett's sphericity test was also applied. The calculated KMO index was 0.911 and a *p*-value < 0.05 was obtained in the Bartlett's sphericity test, which indicated that the dataset including the eleven items of self-efficacy was suitable to be analysed by means of PCA.

The results of the application of the factorial analysis on the self-efficacy dataset are summarised in Table 5 for the two first components. These two principal components explained 80.16% of the total variance. Self-efficacy items 1 and 10 were positively correlated among them since both of them presented high loading values for component number one. Considering the nature of both items (Table 2), it could be stated that component 1 represented the achievement of self-efficacy. The rest of the self-efficacy items, i.e., items 2–9 and 11, presented high loading values in component 2, which means on the one hand that all of them were positively correlated and, on the other hand and considering the nature of these items (Table 3), that component 2 represented the self-efficacy towards learning.

**Table 5.** Results of the factorial analysis on self-efficacy towards the learning of the curricular content related to chemical reactions.

| Self-Efficacy Items | | Component 1 | Component 2 |
|---|---|---|---|
| Achievement self-efficacy | Item 1 | 0.807 | |
| | Item 10 | 0.808 | |
| Learning Self-efficacy | Item 2 | | 0.804 |
| | Item 3 | | 0.781 |
| | Item 4 | | 0.816 |
| | Item 5 | | 0.834 |
| | Item 6 | | 0.803 |
| | Item 7 | | 0.847 |
| | Item 8 | | 0.870 |
| | Item 9 | | 0.713 |
| | Item 11 | | 0.736 |

*3.3. Impact of the Implemented Instruction in Terms of Emotions—Comparison between Control and Experimental Groups*

Figure 2 shows the average scores for the monitored emotions, including positive and negative ones, experienced by the K-9 students included in the sample (control and experimental groups), when learning the curricular content in relation to chemical reactions. As stated above, to measure these emotions, it the mean of each emotion within an appropriate scale from 0 (the lowest frequency) to 10 (the highest frequency) was used.

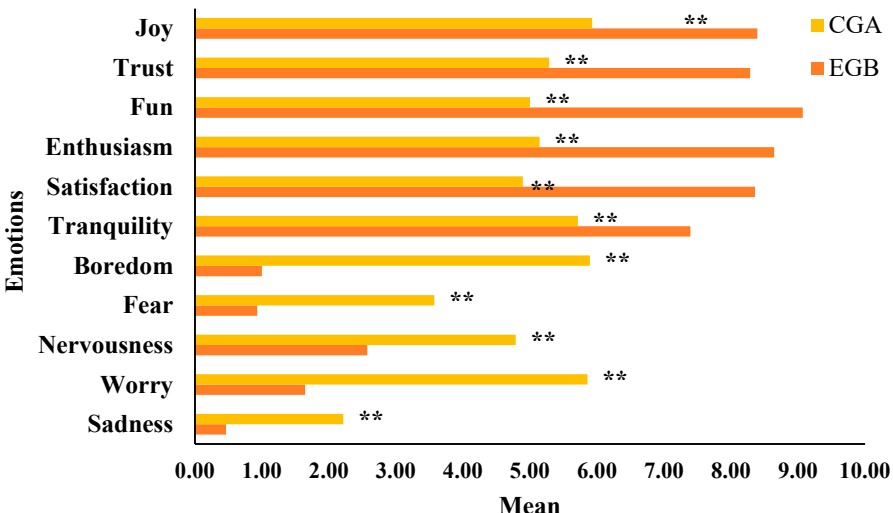

**Figure 2.** Average scores for the emotions experienced by students when learning the curricular content in relation to chemical reactions. Comparison of the control group and experimental group. (** level of significance $p \leq 0.010$ for *t*-Student test).

As can be observed in Figure 2, the average scores in relation to positive emotions for the control group (CGA) ranged between 4.89 (satisfaction) and 5.93 (joy), while in the experimental group (EGB), a significant increase was observed for all the monitored positive emotions, with values above 8 for all of them. In the case of negative emotions, higher values were registered for all students in the control group (CGA) in comparison with the experimental group (EGB), showing statistically significant differences for all the monitored negative emotions.

In addition, the *t*-Student test was used to determine the existence of significant differences between the average scores of emotions, both positive and negative, experienced in the control group and experimental group. As reflected in Figure 2, it can be said that there was a significant increase in the average score values for all the monitored positive emotions experienced by students in the experimental group with an active and innovative strategy when learning chemical reactions, namely for joy ($p = 0.000$), trust ($p = 0.000$), fun ($p = 0.008$), enthusiasm ($p = 0.000$) and satisfaction ($p = 0.000$). On the other hand, the control group whose methodology was traditional magistral expositive lessons experienced a significant increase in the average score values of negative emotions such as boredom ($p = 0.000$), fear ($p = 0.000$), nervousness ($p = 0.000$), worry ($p = 0.000$) and sadness ($p = 0.000$), with respect to the experimental group.

The employed active methodology strategy contributed to the achievement of more effective learning, since a positive affective response was obtained, which is known to help students to learn [18,21]. It was found that students experienced an increased frequency of positive emotions such as satisfaction and a reduction in the frequency of negative emotions when receiving an intervention based on an active methodology. These results are in accordance with studies reported by Brígido et al., (2013) [25].

Other studies found a high occurrence of positive emotions and a low occurrence of negative ones in the case of students in an experimental group receiving an active learning instruction in the framework of a program for the distance learning of science [51]. In the

same line, Sánchez et al., (2017) [40] reported an improved emotional yield and improved cognitive performance when implementing a methodology based on gamification, in comparison with a traditional methodology.

Finally, Sánchez et al., (2018) [52] analysed the emotional responses of prospective teachers toward the learning of science content using different methodologies in the classroom. Specifically, with the gamification and hands-on activities as methodological approaches, students experienced higher positive emotion scores, such as surprise and satisfaction from boredom (static emotion) when the teacher explained science concepts.

### 3.4. Impact of the Implemented Instruction in Terms of Self-Efficacy—Comparison between Control and Experimental Groups

Table 6 shows the comparison in the average scores for self-efficacy beliefs expressed by K-9 students in the considered sample (experimental and control groups) when learning the curricular content in relation to chemical reactions. As stated above, self-efficacy measures were performed within an appropriate scale from 0 (strongly disagree) to 10 (strongly agree).

**Table 6.** Score comparison of self-efficacy beliefs in the control group and the experimental group and the level of significance.

| Items | Self-Efficacy | CGA | | EGB | | Level of Significance |
|---|---|---|---|---|---|---|
| | | Mean | Sd | Mean | Sd | |
| I1 | I can get good results in this topic | 6.68 | 1.105 | 8.43 | 1.136 | <0.010 |
| I2 | I have been able to relate learned concepts about chemical reactions to aspects of daily life | 5.29 | 1.163 | 7.46 | 0.936 | <0.010 |
| I3 | I have been able to understand the chemical reaction concept | 5.57 | 1.031 | 7.86 | 0.956 | <0.010 |
| I4 | I have been able to understand law of mass conservation in the simple processes | 5.64 | 1.055 | 7.71 | 0.838 | <0.010 |
| I5 | I have been able to understand the concepts about physical and chemical change | 6.04 | 0.999 | 9.04 | 1.03 | <0.010 |
| I6 | I have been able to identify a chemical change | 6.36 | 1.222 | 7.82 | 0.793 | <0.010 |
| I7 | I have been able to identify a physical change | 6.57 | 1.424 | 7.96 | 0.905 | <0.010 |
| I8 | I have been able to distinguish chemical change from a physical change | 6.21 | 1.516 | 7.86 | 0.497 | <0.010 |
| I9 | I have been able to read, represent and adjust chemical reactions | 5.89 | 1.141 | 8.14 | 0.693 | <0.010 |
| I10 | If I try hard enough, I will be able to solve tasks | 6.04 | 1.079 | 7.61 | 0.891 | <0.010 |
| I11 | I have been able to relate that I have learned about chemical reactions with hands-on activities outside the classroom | 5.75 | 1.161 | 6.43 | 0.645 | <0.010 |

As reflected in Table 6, the average score for the considered self-efficacy items ranged between 5.29 and 6.68 for the control group and between 6.43 and 9.04 for the experimental group. In the case of the control group, the minimum average score value was collected for item 2 "I have been able to relate learned concepts about chemical reactions to aspects of daily life", which in a way demonstrated that fully significant learning was not achieved with the traditional methodology. The maximum average score values for the control group were registered for item 1 "I can get good results in this topic", followed by item 7 "I have been able to identify a physical change". When comparing the average score values of the control and the experimental groups, statistically significant increases were found for all the measured items, as reflected in Table 6, with the more important increases for item 5 "I have been able to understand the concepts about physical and chemical change", followed by item 3 "I have been able to understand the chemical reaction concept" and item 9 "I have been able to read, represent and adjust chemical reactions".

The statistical *t*-Student test was employed to check the existence of statistically significant differences between self-efficacy beliefs in the control and experimental groups. In accordance with the results collected in Table 6, it can be stated that there was a significant increase in the average score of self-efficacy experienced by students in the experimental group with an active and innovative strategy when learning the chemical reactions in all proposed items. On the other hand, the control group whose methodology was traditional magistral expositive lessons experienced a significant decrease in the average score compared with the experimental group.

It has been reported in the literature [51–53] that the promotion of self-efficacy beliefs results in increased motivation, performance and ability to develop their academic tasks amongst students. It has also been reported that students develop more capabilities to solve complex problems [45].

## 4. Conclusions

In the frame of the current research, it has been proved that students feel and react in different ways in terms of emotions and self-efficacy when the teacher employs an active methodology or a traditional one in the classroom. The curricular content in relation to chemical reactions was selected as a case study.

According to the results, the implementation of an active and innovative methodology with the experimental group had a positive effect in terms of an increased occurrence of positive emotions. There was a significant increase in the average scores in relation to positive emotions such as joy, trust, enthusiasm, fun, tranquility and satisfaction in the experimental group compared to the control group. On the other hand, a significant decrease in the average scores in relation to negative emotions such as boredom, fear, nervousness, worry and sadness were given by the students in the experimental group.

With respect to self-efficacy beliefs, it can be stated that the active learning methodology was proven to be effective since an increase in self-efficacy belief scores toward the learning of chemical reactions was registered with a methodology based on inquiry through hands-on activities.

These results should encourage teachers to pay special attention when scheduling their teaching activities. It is necessary that the teacher reflects and develops innovative and motivating methodologies in the teaching/learning process. The monitorisation of emotions and self-efficacy beliefs when learning the content might assist the teacher in the selection of strategies that promote a positive learning climate.

**Author Contributions:** Conceptualization, M.A.D.-A., J.S.-M., D.A.-R. and F.C.-C.; methodology, M.A.D.-A., J.S.-M., D.A.-R. and F.C.-C.; software, M.A.D.-A. and D.A.-R.; validation, M.A.D.-A., J.S.-M., D.A.-R. and F.C.-C.; formal analysis, M.A.D.-A. and D.A.-R.; investigation, M.A.D.-A., J.S.-M., D.A.-R. and F.C.-C.; resources, J.S.-M. and F.C.-C.; data curation, M.A.D.-A.; writing—original draft preparation, M.A.D.-A., J.S.-M., D.A.-R. and F.C.-C.; writing—review and editing, M.A.D.-A., J.S.-M., D.A.-R. and F.C.-C.; visualization, F.C.-C.; supervision, F.C.-C. and D.A.-R.; project administration, J.S.-M. and F.C.-C.; funding acquisition, J.S.-M. and F.C.-C. All authors have read and agreed to the published version of the manuscript.

**Funding:** This work was funded by Junta de Extremadura and "ERDF A way of making Europe", through the Project GR21047 and IB18004, as well as by the Spanish Ministry of Science and Innovation and the State Research Agency through grant PID2020-115214RB-I00 funded by MCIN/AEI/10.13039/501100011033.

**Institutional Review Board Statement:** Not applicable. This is a non-interventional study. All participants were informed about their assured anonymity, about why the research was being conducted and how their data were going to be used. According to Spanish legislation (Organic Law 15/1999), no further ethical approval is required.

**Informed Consent Statement:** Informed consent was obtained from all subjects involved in the study.

**Data Availability Statement:** Not applicable.

**Conflicts of Interest:** The authors declare no conflict of interest.

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
