# Peer review of "Impact of an Active Learning Methodology on Students’ Emotions and Self-Efficacy Beliefs towards the Learning of Chemical Reactions—The Case of Secondary Education Students"

_education, doi:10.3390/educsci12050347_

Round 1

Reviewer 1 Report

This article presents a comparative study of the role of active learning in the affective domain including student emotions and self-efficacy.  While the results showed positive outcomes for the treatment vs. control group, I have some significant concerns with the study:

  • The student sample size here is small and is not gender balanced. How do the authors see this contributing to the outcome?
  • As the sample size is small, to what extent did the authors categorize the similarity and differences of the treatment and control groups?
  • While the survey instrument was purported to relate to self-concept and self-efficacy, I did not find it so. For example, a statement like: “If I try hard enough, I will succeed” is fine, but other statements are related to educational outcomes rather than statements of self-concept.  For example: “I have been able to understand the concepts about physical and chemical change”.  How is that related to self-concept?  The student is merely responding as to whether they have understood these concepts, not their self-concept around this understanding.  Given the instruments available to measure self-efficacy and self-concept in the literature, I am surprised that one of these was not used here. 

Author Response

This article presents a comparative study of the role of active learning in the affective domain including student emotions and self-efficacy.  While the results showed positive outcomes for the treatment vs. control group, I have some significant concerns with the study:

Point 1: The student sample size here is small and is not gender balanced. How do the authors see this contributing to the outcome?

Response 1: The sample size is considered normal for a compulsory secondary school in Spain. The group formation in the high schools is made randomly, as stated in the article, namely students are divided in alphabetical order of surnames, thus it is logical that it might result in slightly unbalanced group regarding the gender. Nevertheless, the global number of boys and girls is around 50%. However, gender studies have not been performed, since it is not among the objectives of the current study.

Point 2: As the sample size is small, to what extent did the authors categorize the similarity and differences of the treatment and control groups?

Response 2: As detailed in the article (paragraph 2.4. Data analysis) “Previously, it has been tested that the sample was homogeneous.” This was checked through a questionnaire about emotions and self-efficacy and it was found that differences did not exist between the experimental and the control group, with regard to emotions and self-efficacy towards the considered curricular content. This is consequence of the followed criterium for group formation, which as stated above, was performed alphabetically.

Point 3: While the survey instrument was purported to relate to self-concept and self-efficacy, I did not find it so. For example, a statement like: “If I try hard enough, I will succeed” is fine, but other statements are related to educational outcomes rather than statements of self-concept.  For example: “I have been able to understand the concepts about physical and chemical change”.  How is that related to self-concept?  The student is merely responding as to whether they have understood these concepts, not their self-concept around this understanding.  Given the instruments available to measure self-efficacy and self-concept in the literature, I am surprised that one of these was not used here. 

Response 3: We agree with the reviewer in this concern. Some of the proposed questions are more focused to measure the perception of students’ self-efficacy in relation with the curricular content treated in the current research. This makes the instrument too specific, that is the reason why we have not taken directly an instrument from the literature. In fact, it has been adapted from the instrument of reference 42, which belongs to our research group. The objective of its application was to check their perception after the instruction was performed in both groups.

Additionally, as it can be seen in Table 5, PCA is able to distinguish between both types of questions in the self-efficacy questionnaire, and two categories are established by the authors “achievement self-efficacy” and “learning self-efficacy”. This strongly supports the comment of the reviewer.

Reviewer 2 Report

English needs work - please work with an English editor or native speaker.

What is the third level of compulsory education? What does K-9 that mean? Level 9? Grade 9? How old are the typical students in this group? 16? This is finally stated in the material and methods section, but should also be mentioned earlier.

For Figure 1, it is assumed that the author(s) created the graphics and there is no need to state “Own elaboration”.

For other units, how did the two groups of students compare? Is it the same teacher for both groups? 

The experiences/activities are unclear – more information is needed. The only one that is clear is the “change of state; the melting of ice”. I am assuming that the way to modify the speed of a chemical reaction is by changing the temperature (could be addition of a catalyst), but how does the oxidation of an apple connect with this?

Was the questionnaire given all at once at the end of all lessons or parts along the way? How would students remember emotions from the first day after many days?

The statistical analysis values need additional context  - e.g. at what value is the Kolmogorov-Smirnov normality test p number meaningful – and what does it signify?

What set of questions does Table 5 correspond to? The new questions or ones from someone else’s set of questions? What were the other questions? (it would be helpful if the information in table 5 were closer to Table 3 so that the analysis of questions could be more easily related to the questions themselves). This is done much better in Table 6.

What type of statistical test was used to determine whether student responses in Figure 2 were significant? If it is the t-test, that should be mentioned before mentioning the level of significance.

The last paragraph before the conclusions make it sound as if the performance of the students on some sort of assessment should have been done and included.  How did the assessment scores of the two groups compare? If there was no difference, then the efficacy does not matter. The assessment results should be included as well. This is also implied in the abstract, yet no information is provided.

25 of the 53 references are older than 10 years.

Author Response

Point 1: English needs work - please work with an English editor or native speaker.

Response 1: Thanks to the reviewer for the comment. Considering the reviewer’s advice, the use of English has been revised throughout the article.

Point 2: What is the third level of compulsory education? What does K-9 that mean? Level 9? Grade 9? How old are the typical students in this group? 16? This is finally stated in the material and methods section, but should also be mentioned earlier.

Response 2: K-9 refers to the international standard. Considering the reviewer comment, it has been clarified in the article the first time that K-9 appears (page 3; section 2. Material and methods; section 2.1. Sample). We have selected this course denomination, to make the article more accessible to an international audience. K-9 corresponds to the third level of compulsory secondary education in the Spanish educational system, with ages of students ranging between 14 and 15 years old. In fact, this is explained in the article (section 2. Material and methods; 2.1. Sample) “The considered sample consists of 56 students of 3rd year compulsory secondary education, i.e. K-9 students according to international standards, (ages ranged: 14-15 years old)…”

Point 3: For Figure 1, it is assumed that the author(s) created the graphics and there is no need to state “Own elaboration”.

Response 3: We agree with the reviewer, and it has been deleted.

Point 4: For other units, how did the two groups of students compare? Is it the same teacher for both groups? 

Response 4: As detailed in the article (section 2.4. Data analysis) “Previously, it has been tested that the sample was homogeneous.” This was checked through a questionnaire about emotions and self-efficacy and it was found that differences did not exist between the experimental and the control group, with regard to emotions and self-efficacy towards the considered curricular content. This is consequence of the followed criterium for group formation, which as stated above, was performed alphabetically.

Regarding the teacher, the instruction was performed by the same teacher in both groups, who was their normal teacher. Additionally, in the experimental group the teacher was assisted by one of the researchers of this article, especially for experiments performance sessions.

Point 5: The experiences/activities are unclear – more information is needed. The only one that is clear is the “change of state; the melting of ice”. I am assuming that the way to modify the speed of a chemical reaction is by changing the temperature (could be addition of a catalyst), but how does the oxidation of an apple connect with this?

Response 5: The objective of the second session described in the article, was to make students to relate macroscopic changes (appearance of brownish color) with chemical changes. Thus, in that case, there was no need to modify the speed of reactions. Regarding the third session, the apple is exposed to oxygen in three parts under different conditions: part number one, completely exposed; part number two, covered with kitchen plastic; and part number three, sprinkled with lemon juice. The employed different conditions turn in different speed for the oxidation reaction. This has been rewritten and clarified in the article, according to the reviewer comment.

Point 6: Was the questionnaire given all at once at the end of all lessons or parts along the way? How would students remember emotions from the first day after many days?

Response 6: As detailed in the article (paragraph 2.4. Data analysis) “Previously, it has been tested that the sample was homogeneous.” This was checked through a questionnaire about emotions and self-efficacy and it was found that differences did not exist between the experimental and the control group, with regard to emotions and self-efficacy towards the considered curricular content. Then, the instruction was performed and at the end of the third session the questionnaire was implemented, thus there was no chance to forget the experienced emotions. This has been clarified at the beginning of section “2.4. Data analysis”, taking into account the reviewer comment: “Students employed an average time of 30 minutes to complete the questionnaire, and they were implemented immediately after the third session.”

Point 7: The statistical analysis values need additional context  - e.g. at what value is the Kolmogorov-Smirnov normality test p number meaningful – and what does it signify?

Response 7: As stated in the article: “… the Kolmogorov-Smirnov normality test (p> .050) establishing that the sample in conformity with the requirements of normally distributed. In this analysis a confidence level of 95% was considered.” In the case of obtaining a p-value in Kolmogorov-Smirnov normality test below 0.050 the sample would be considered non-parametric, but this is not the case.

Point 8: What set of questions does Table 5 correspond to? The new questions or ones from someone else’s set of questions? What were the other questions? (it would be helpful if the information in table 5 were closer to Table 3 so that the analysis of questions could be more easily related to the questions themselves). This is done much better in Table 6.

Response 8: Items of Table 5 are the same ones as those of Table 3, they correspond to the third section of the questionnaire. Table 5 summarizes, in fact, the results of the analysis of items from table 3. Table 3 is focused in the description of the instrument, while Table 5 is focused on the results of data analysis, we consider that including the description of each item again in Table 5 supposes duplication of information.

Point 9: What type of statistical test was used to determine whether student responses in Figure 2 were significant? If it is the t-test, that should be mentioned before mentioning the level of significance.

Response 9: We agree with the reviewer comment, and considering it, caption for Figure 2 has been changed to: “Figure 2. Average scores for emotions experienced to students when learning chemical reactions curricular content. Comparison of control group and experimental group. (** level of significance p≤.010 for t-Student test).”

Point 10: The last paragraph before the conclusions make it sound as if the performance of the students on some sort of assessment should have been done and included.  How did the assessment scores of the two groups compare? If there was no difference, then the efficacy does not matter. The assessment results should be included as well. This is also implied in the abstract, yet no information is provided.

Response 10: This research is focused on the effect of the employed methodology on emotions and self-efficacy. Students performance is not included since it does not match with the research objectives. Conclusions have been rewritten in the revised version of the manuscript, considering this comment, and also for grammatical correction.

Point 11: 25 of the 53 references are older than 10 years.

Response 11: We decided to include in the reference list pioneering research reports about the influence of affective domain in science education.

The study of the affective domain in science education took off at the beginning of the 21st century. We consider that it is an emerging line of research and from our point of view, we consider it necessary to cite the main references.

However, if the Editor considers that prior to publication, if applicable, it is necessary to delete some of these references, we have no problem to do it. 

Round 2

Reviewer 1 Report

The authors have responded to my comments.  They should indicate whether this study received IRB or ethics approval at their institution.

Author Response

Regarding the information about Ethics Comittee or Institutional Review Board approval, we have checked with our university (University of Extremadura, Spain) and we do not need any special approval due to the fact that the nature of this piece of research does not involve identified patients or similar research objects. The data collection from individuals was performed in accordance with the ethical standards of the 1964 Helsinki declaration. Informed consent was obtained from all individuals and their anonymity was guaranteed by double-blind procedure. The corresponding ethical committe (Comisión de Bioética y Bioseguridad, Universidad de Extremadura, Spain) includes this kind of studies as those with no need for specific approval procedure. All information about the bioethics and ethics in research activity at University of Extremadura can be retrieved from http://investigalia.unex.es [Accessed on 04/22/2022].Consequently, we ask for including the following statements in our current manuscript:

Institutional Review Board Statement: Not applicable.

Informed Consent Statement: Informed consent was obtained from all subjects involved in the study.

Reviewer 2 Report

Was this research pre-approved by an ethics committee?

25 of the 53 references are more than 10 years old.

English language word usage and punctuation needs to be improved.

Author Response

Was this research pre-approved by an ethics committee?

Regarding the information about Ethics Comittee or Institutional Review Board approval, we have checked with our university (University of Extremadura, Spain) and we do not need any special approval due to the fact that the nature of this piece of research does not involve identified patients or similar research objects. The data collection from individuals was performed in accordance with the ethical standards of the 1964 Helsinki declaration. Informed consent was obtained from all individuals and their anonymity was guaranteed by double-blind procedure. The corresponding ethical committe (Comisión de Bioética y Bioseguridad, Universidad de Extremadura, Spain) includes this kind of studies as those with no need for specific approval procedure. All information about the bioethics and ethics in research activity at University of Extremadura can be retrieved from http://investigalia.unex.es [Accessed on 04/22/2022].Consequently, we ask for including the following statements in our current manuscript:
Institutional Review Board Statement: Not applicable.

Informed Consent Statement: Informed consent was obtained from all subjects involved in the study.

25 of the 53 references are more than 10 years old.

We decided to include in the reference list pioneering research reports about the influence of affective domain in science education.

The study of the affective domain in science education took off at the beginning of the 21st century. We consider that it is an emerging line of research and from our point of view, we consider it necessary to cite the main references.

However, if the Editor considers that prior to publication, if applicable, it is necessary to delete some of these references, we have no problem to do it.

English language word usage and punctuation needs to be improved.

Considering the reviewer’s advice, the use of English was revised by a native English-speaker throughout the article when we prepared the prior revised version. All changes were done with the “track changes” option activated in “word” text processor. However, considering the reviewer comment, the English word usage and punctuation have been revised again. 

This manuscript is a resubmission of an earlier submission. The following is a list of the peer review reports and author responses from that submission.